# Impact of Virtual vs. In-Person School on Children Meeting the 24-h Movement Guidelines during the COVID-19 Pandemic

**DOI:** 10.3390/ijerph191811211

**Published:** 2022-09-07

**Authors:** Christopher D. Pfledderer, Michael W. Beets, Sarah Burkart, Elizabeth L. Adams, Robert Glenn Weaver, Xuanxuan Zhu, Bridget Armstrong

**Affiliations:** Arnold School of Public Health, University of South Carolina, Columbia, SC 29208, USA

**Keywords:** school-based physical activity, youth physical activity promotion, children, youth

## Abstract

The pandemic mitigation strategy of closing schools, while necessary, may have unintentionally impacted children’s moderate-to-vigorous physical activity (MVPA), sleep, and time spent watching screens. In some locations, schools used hybrid attendance models, with some days during the week requiring in-person and others virtual attendance. This scenario offers an opportunity to evaluate the impact of attending in-person school on meeting the 24-h movement guidelines. Children (N = 690, 50% girls, K–5th) wore wrist-placed accelerometers for 14 days during October/November 2020. Parents completed daily reports on child time spent on screens and time spent on screens for school. The schools’ schedule was learning for 2 days/week in-person and 3 days/week virtually. Using only weekdays (M–F), the 24-h movement behaviors were classified, and the probability of meeting all three was compared between in-person vs. virtual learning and across grades. Data for 4956 weekdays (avg. 7 d/child) were collected. In-person school was associated with a greater proportion (OR = 1.70, 95% CI: 1.33–2.18) of days that children were meeting the 24-h movement guidelines compared to virtual school across all grades. Students were more likely to meet the screen time (OR = 9.14, 95% CI: 7.05–11.83) and MVPA (OR = 1.50, 95% CI: 1.25–1.80) guidelines and less likely to meet the sleep (OR = 0.73, 95% CI: 0.62–0.86) guidelines on the in-person compared to the virtual school days. Structured environments, such as school, have a protective effect on children’s movement behaviors, especially physical activity and screen time.

## 1. Background

The 24-h movement guidelines for youth outline an optimal composition of movement behaviors for the 24-h day. For youth aged 5–13 years, the guidelines recommend 60 min per day of moderate to vigorous physical activity (MVPA), 9–11 h of sleep per night, and no more than 2 h per day of recreational screen time [1]. The guidelines have been adopted by several countries and agencies for multiple age groups [2,3,4,5], and many researchers have integrated the guidelines in their research designs for both observational and intervention-based research [6]. Meeting the youth 24-h movement guidelines is associated with lower adiposity [7], higher fitness [8], better dietary patterns [9], mental health [10], and health-related quality of life [11]. While the 24-h movement guidelines are important for children to achieve daily, the pandemic mitigation strategy of closing schools to slow the spread of the novel SARS-2 COVID virus may have had an unintended impact on children’s MVPA, sleep, and time spent watching screens.

At the height of the pandemic, multiple mitigation strategies were adopted by K–12 schools. These included the following: (1) schools completely shut down, with all students receiving online instructions, (2) alternating school days (hybrid), in which half of the students receive in-person instructions some days of the week and virtual on others, and (3) families opting to have their children receive in-person or online instructions [12]. The positive impact that schools have on the movement behaviors of youth is well documented [13,14], but such findings typically compare weekdays, when children attend school, to weekend days. There is a paucity of research comparing the movement behaviors between children attending in-person and virtual school, which could provide insights about the impact of in-person school on youth health behaviors. Further, to date, no study has utilized within-person analyses to compare the movement behaviors between days when a child attends an in-person school versus days when that same child attends a virtual school. Previous studies report that children engaged in fewer health-promoting behaviors during the pandemic, when schools were partially or fully closed, compared to times either before or after the pandemic [15]. The limitation of these studies is the reliance on repeated cross-sections of children at differing time points or data collected during differing seasons or weather. Objective data collected when children attend hybrid schooling can provide unique insights into the role that schools play in promoting healthful behaviors, particularly during a pandemic, and unique insights into how the mode of education delivery may impact healthful behaviors as well.

The pandemic mitigation strategy of hybrid schooling offers an unprecedented natural experiment with the ability to examine the influence of schools on meeting the 24-h movement guidelines. The purpose of this study was to evaluate the impact of attending an in-person versus a virtual school on meeting the 24-h movement by comparing days when children attended an in-person school to days when the same children attended a virtual school during a period of hybrid instruction in K–5 schools in the fall of 2020.

## 2. Methods

School and Child Sampling. Data for this study came from a longitudinal cohort study, which measured children’s obesogenic behaviors (i.e., physical activity, sedentary time, sleep, screen time, and diet). Only measures during the COVID-19 pandemic collected in October and November 2020 were used for this analysis. The participants were recruited via two neighboring school districts in the southeastern United States, which served K–5th grades. No exclusion criteria were used prior to the recruitment. All of the study procedures were approved by the university’s Institutional Review Board (IRB#Pro00080382).

Procedures—physical activity, sleep, and screen time. The objectively measured MVPA, objectively measured sleep, and daily collection of parent-reported child screen times were utilized for this study. The children were asked to wear an Actigraph GT9X accelerometer on their non-dominant wrist for 24 h per day for 14 days in October or November 2020. The parents received a brief online survey each night of the 14-day wear period and were asked to provide information about their child’s day, including whether or not they attended school virtually or in-person, the amount of time their child watched or used screens for educational and non-educational purposes, and bed and wake times.

Accelerometry. The Actigraph GT9X accelerometers were initialized and downloaded using the Actilife software (version 6.13.4, Pensacola, FL, USA). The accelerometers were initialized to record data at a frequency of 30 Hz and began data collection at 7:00 AM on the day preceding the earliest device delivery. Stop time was not used. The idle sleep mode was enabled to preserve battery life, and the display was turned off to limit the distractions for children while attending school. The data were downloaded and saved in a raw format as GT3X files and converted to .csv files for processing. The raw .csv files were processed using the GGIR package (version 2.6-0) [16] in R (Version 4.1.2; R Foundation for Statistical Computing; Vienna, Austria). The time spent in physical activity intensity categories was determined by using intensity thresholds described by Hildebrand et al. [17], as these are the only cut-points validated for use with the GGIR package [18,19,20]. The sleep estimates were guided by the use of parent-reported bed and wake times. On nights when the parents provided bed and wake times, the advanced sleep log option in GGIR was used to guide the detection of the sleep period. When parent-reported sleep logs were not available, the default algorithm within the GGIR was used to identify the sleep period [21]. Valid accelerometry data for a single day were a minimum of 16 h of wear. The participants were included in the analyses if they had at least one valid day for in-person and one valid day for virtual school.

24-h Movement Behavior Classification. Each movement behavior for each valid day was dichotomized into meet (1) or not meet (0), based upon the Canadian guidelines for children 5–12 years old [1]. The decision to classify each day as meeting or not meeting the guidelines was made to examine the probability of meeting the guidelines on days when attending in-person school versus virtual school.

Physical activity: an accumulation of ≥60 min per day of MVPASleep: 9–11 h of sleep per nightScreen time (sedentary behavior): ≤2 h per day of recreational screen time

Statistical Analyses. Initially, the descriptive means and standard deviations were calculated for all of the variables. For the primary analyses of interest, multi-level logit models were used to examine the probability of meeting each of the movement behaviors individually and all three guidelines on days when the children attended in-person school versus virtual school. Only weekdays (M–F) were used in the analyses. The logit models included the following: the main effect of school type (in-person vs. virtual), grade, school type-x-grade interaction, sex, race, parent education, and household income. The models were nested with multiple days within children and children within schools. The same modeling approach with the continuous outcomes (minutes/day) of MVPA, sleep, and screen time was used to examine the impact of school type on these behaviors. For the logit and continuous models, evaluating meeting all 3 of the movement behavior guidelines and the screen time guidelines, an additional covariate of the number of minutes per day using screens for educational purposes was included in the models. All analyses were performed with Stata v.16.1 statistical software package (College Station, TX, USA).

## 3. Results

Descriptive Characteristics. The descriptive characteristics of the sample and the number of days of valid accelerometer wear for in-person and virtual school days are presented in Table 1. The participants (n = 690, 50% female, 60% white, and 42% at or below the 200% federal poverty level) accumulated a total of 4956 valid accelerometer wear days. Of those, 1685 days (33.9%, mean/participant = 3.6 days) were spent attending in-person school and 3271 days (66.1%, mean/participant = 4.8 days) were spent attending virtual school. The accelerometer wear time and compliance did not differ across grade levels.

24-h Movement Behaviors. The odds of meeting the movement behavior guidelines by school type (in-person vs. virtual) are presented in Table 2. Overall, the participants were more likely to meet all three movement behaviors and the MVPA and screen time guidelines on days when attending in-person school compared to days attending virtual school. Conversely, the participants were less likely to meet the sleep guidelines on days that they were attending in-person school. For meeting all three guidelines, MVPA, and sleep, there was a clear and statistically significant interaction between attending in-person school and grades, such that as a grade increased, the children were more likely to meet the guidelines when attending in-person school. The average minutes spent in each movement behavior while attending in-person versus virtual school as well as the interaction estimates for school type-x-grade are displayed in Figure 1. The proportion of days where the 24-h movement behavior guidelines were met when children attended in-person versus virtual school is shown in Figure 2. Across all grades, the proportion of children that meet all three guidelines and the MVPA and screen time guidelines separately was higher when attending in-person compared to virtual school. The opposite scenario was observed for the proportion of children meeting the sleep guidelines, such that the proportion of children meeting the sleep guidelines was higher when attending virtual school compared to in-person school.

## 4. Discussion

This study evaluated the influence of attending in-person versus virtual school on meeting the 24-h movement guidelines in a sample of children who experienced a hybrid learning scheme during October and November 2020, when pandemic mitigation strategies were common [12]. On days when children attended school in-person, they had higher odds of meeting all three 24-h movement behaviors and higher odds of meeting the physical activity and screen time guidelines individually. It was also found that as the age (grade) increased, the protective effect of attending in-person school on meeting the 24-h movement behaviors increased, even though both the MVPA and sleep duration declined and the screen time trended upward. This study is one of the first to demonstrate the effects of virtual versus in-person school attendance on device-based 24-h movement behaviors in children and one of the first to conduct within-person analyses comparing the movement behaviors between days when children are attending virtual versus in-person school, which further demonstrates the positive impact that structure might have on children’s health behaviors, particularly for older elementary-age children.

These findings are consistent with the Structured Days Hypothesis [13], which states that children are more likely to engage in healthful behaviors when attending structured environments, such as school. The reason for this is the purposeful, segmented, restrictive, and compulsory components embedded within a school day that lead to children engaging in more healthful behaviors. On days of the week when children attend virtual school at home, it is likely that there will be fewer opportunities to engage in physical activity and fewer limits on the time a child can recreationally watch screens. The data presented here support that higher activity levels and lower screen times are seen on days spent attending in-person school. If schools continue to adopt a hybrid structure, future studies should explore how the virtual school environment could be adapted to promote healthy 24-h movement behaviors the same way in-person schools have been shown to do.

The importance of attending in-person school is further supported by the increase in the proportion of days meeting the guidelines for all three behaviors, MVPA, and screens for older children. It is well documented that children’s MVPA decreases and screen time increases from childhood to adolescence [22,23,24]. A similar pattern is observed in this sample. When the older children attended in-person school, they engaged in more MVPA and had less screen time compared to days spent attending virtual school. This suggests that the health benefits of attending school may be more pronounced for older children. This may be driven by the greater amounts of autonomy that older children have compared to younger children [25], or be developmentally induced with younger children wanting to play more than older children [26]. Despite the cause, attending in-person school appears to promote more healthful activities and screen time behaviors compared to when children are at home attending virtual school. This has important implications for schools and how their structures can help minimize the decline in activity and the increase in screen time as children age.

### 4.1. Implications and Future Directions

This study provides a potential road map for future investigations of school-based movement behavior promotions. First, school-based interventions have been shown to only be marginally effective at increasing healthy movement behaviors [27,28,29], but our study provides evidence that in-person school might protect against declines in physical activity and increases in screen time that occur when children are in a less structured environment. Future school-based studies should investigate the specific factors of the school environment that contribute to the promotion of health behaviors and design interventions that maximize the impact of these factors. Another point to consider is the fact that virtual school attendance was often mandatory (forced choice), which does not accurately represent a typical behavioral intervention, although it did provide an opportunity to investigate the effects of a natural experiment. Researchers should continue to leverage the opportunity to investigate these types of natural experiments by identifying when changes to the infrastructure, policies, or practices may occur or have occurred. Our study also showed that the in-person school environment had a particularly positive impact on older children’s movement behaviors, but questions remain as to why that is the case. Future studies should disentangle the age-related factors contributing to this phenomenon. Finally, as schools plan for the future, which may include providing hybrid and virtual options for education delivery, researchers should continue to investigate ways in which the virtual school environment may be tailored to better promote health behaviors for the children utilizing this option [30].

### 4.2. Strengths and Limitations

The strengths of this study are the natural experiment of alternative virtual vs. in-person school days during the same two-week assessment period, the use of objective measures for physical activity and sleep, daily reports of screen time use, and the large and diverse sample of children. This study was not without limitations, which should be considered when interpreting the results. Even though this study can be considered a natural experiment, the virtual school attendance was mandatory and not a conventional teaching arrangement, which limits the external validity of the research findings. While this is standard practice with most studies investigating the 24-h movement behaviors among youth [6], screen time measures rely on parent-report surveys, which are not as rigorous as the objective measures for screen time. This was also a cross-sectional study without any control group comparisons, although we did conduct within-person analyses (where children were acting as their own controls) and controlled for multiple demographic covariates to account for some endogeneity. While our sample demographics are similar to those of the southeastern United States, the results may only be generalizable to that area. Finally, we did not collect information on the school schedules for the in-person and virtual school days; therefore, we cannot comment on what contextual components of the schools were present or not present when comparing in-person versus virtual schools.

## 5. Conclusions

In conclusion, this study demonstrates that attending an in-person school has an overall positive impact on meeting all three of the movement guidelines and the MVPA and screen time guidelines separately. This positive impact appears to be greater for older children. These findings provide evidence that attending an in-person school plays an important role in promoting healthful behaviors in children and helps to explain why the health behaviors of children declined during the pandemic, which likely led to the dramatic increases in overweight and obesity observed among youth [31,32]. Future studies should explore the specific factors associated with attending school in-person that contribute to the promotion of movement behaviors, including physical activity and screen time; further investigate the age-related differences in both the 24-h movement behaviors and the influence of in-person school on the movement behaviors; and investigate ways in which virtual schools could better protect against declines in meeting the 24-h movement guidelines.

## Figures and Tables

**Figure 1 ijerph-19-11211-f001:**
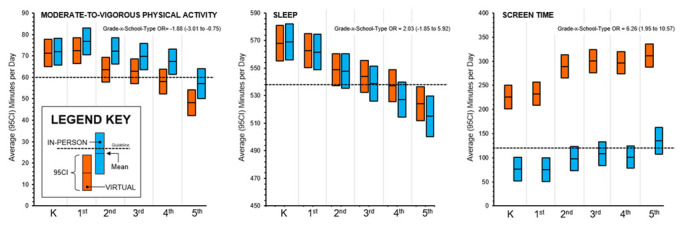
Average (95% confidence intervals) minutes spent in each movement behavior while attending virtual (orange) or in-person (blue) school for elementary-age children during the same 14-day measurement period in October or November 2020.

**Figure 2 ijerph-19-11211-f002:**
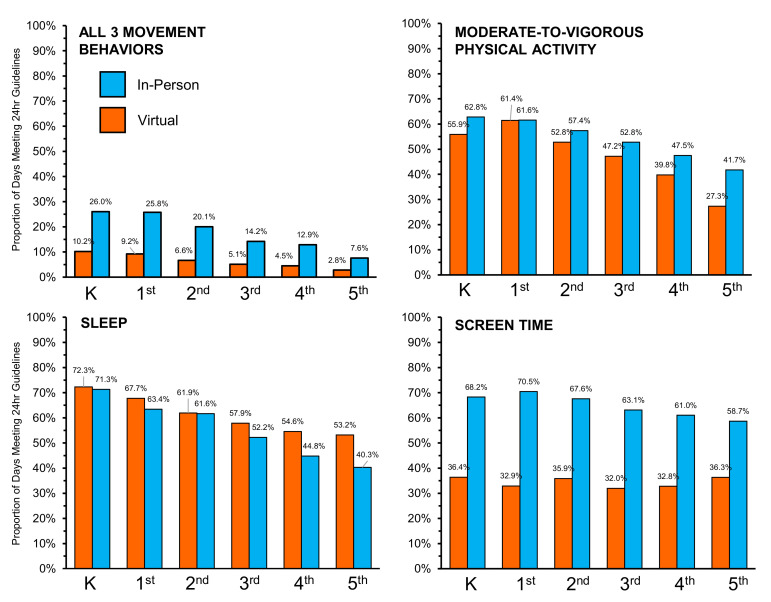
Proportion of days that the movement behavior guidelines were met during virtual (orange) and in-person (blue) school for elementary-age children during the same 14-day measurement period in October or November 2020.

**Table 1 ijerph-19-11211-t001:** Descriptive characteristics of the sample (N = 690).

Characteristic	
Girls (%)	50%
Grade (*n*)	
K	94
1st	77
2nd	127
3rd	143
4th	157
5th	92
Parent Education (N)	
High school or less	37
Some college	152
2 or 4 year degree	306
Graduate degree	195
Household Income	
<$40,000	151
$40,000 to <$80,000	209
>$80,000	331
% At or Below 200% Federal Poverty Level	42%
Race/Ethnicity (%)	
Black	30%
Other	10%
White	60%
Accelerometer Wear	
Total Days	4956
Avg Days/Participant	7
School Type (# days)	
In-Person	1685
Avg days/participant	3.6
Virtual	3271
Avg days/participant	4.8

**Table 2 ijerph-19-11211-t002:** Odds of meeting the movement behavior guidelines while attending in-person compared to virtual school for elementary-age children during the same 14-day measurement period in October or November 2020.

	Estimates	
24-h Movement Behavior	OR	(95% CI)
All 3 Movement Behaviors	1.70	(1.33, 2.18)
Moderate-to-Vigorous Physical Activity	1.50	(1.25, 1.80)
Sleep	0.73	(0.62, 0.86)
Screentime	9.14	(7.06, 11.83)

Note: Model controls for gender, race, parent education, and household income.

## Data Availability

The data presented in this study are available upon request from the corresponding author. The data are not publicly available due to restrictions based on privacy and research ethics.

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
