# Peer review of "Impact of Virtual vs. In-Person School on Children Meeting the 24-h Movement Guidelines during the COVID-19 Pandemic"

_ijerph, 2022, doi:10.3390/ijerph191811211_

Round 1

Reviewer 1 Report

This study examines differences in children's physical activity, sleep, and screen time during vitual vs. in-person learning days during the COVID-19 pandemic. The paper is concise, clear, and has compelling findings; it will be a welcome addition to the literature. I had only a few comments.   Introduction -should clarify which guidelines they're referring to (national guidelines for the US?) -introduction is generally very well written and clearly makes a case for the study   Methods/Results -the authors mentioned the accelerometer cut points were based on Hildebrand et al. - a bit more information on the validity of the cut points would be helpful. As yet there are no globally accepted cut points for children, so some background on choosing these cut points would be helpful -since the authors have a large enough sample, they should run a formal interaction analysis on differences in movement behaviors across ages in in-person vs. virtual school - I suspect it would be significant -did accelerometer wear time/compliance differ across grades? 

Author Response

Dear Reviewer,

Please see attached for responses to your most helpful comments. We have also highlighted changes to the manuscript in yellow in the manuscript file.

Thank you.

Reviewer 2 Report

This study examined the influence of attending in-person versus virtual school on meeting 24-hour movement guidelines, which is quite interesting and valuable.

1 The 24-hour movement guidelines are relatively unfamiliar to international readers. In the background introduction, authors are advised to include a more detailed introduction, such as the purpose, the actual effect, and whether there is relevant research support.

2 Demographic characteristics of the sample should be reported.

3 The information presented in the tables and figures should be properly explained, especially the reason for the result.The explanation currently in the discussion does not seem to be comprehensive and complete enough, e.g. In the last figure of the Figure 2, why does screen time in virtual school is much lower than in-person school. This conclusion is likely to be highly inconsistent with our expectations, at least in my culture.

4 The virtual school under the epidemic is a forced choice, not a conventional teaching arrangement, which limits the external validity of the research. In research discussions, authors should explore more deeply the value of the research findings and how this research can be applied in regular teaching.

Author Response

(The authors gave the same response as above.)

Round 2

Reviewer 2 Report

The authors have made revisions very efficiently. I think the manuscript has been sufficiently improved to warrant publication.